# Effectiveness of Bacteriophages against Biofilm-Forming Shiga-Toxigenic *Escherichia coli* In Vitro and on Food-Contact Surfaces

**DOI:** 10.3390/foods12142787

**Published:** 2023-07-22

**Authors:** Divya Jaroni, Pushpinder Kaur Litt, Punya Bule, Kaylee Rumbaugh

**Affiliations:** Department of Animal and Food Sciences, and Food and Agricultural Products Center, Oklahoma State University, N. Monroe Street, Stillwater, OK 74078, USA

**Keywords:** bacteriophages, Shiga-toxigenic *E. coli*, biocontrol, biofilms, food-contact surfaces

## Abstract

(1) Background: Formation of biofilms on food-contact surfaces by Shiga-toxigenic *Escherichia coli* (STEC) can pose a significant challenge to the food industry, making conventional control methods insufficient. Targeted use of bacteriophages to disrupt these biofilms could reduce this problem. Previously isolated and characterized bacteriophages (*n* = 52) were evaluated against STEC biofilms in vitro and on food-contact surfaces. (2) Methods: Phage treatments (9 logs PFU/mL) in phosphate-buffered saline were used individually or as cocktails. Biofilms of STEC (O157, O26, O45, O103, O111, O121, and O145) were formed in 96-well micro-titer plates (7 logs CFU/mL; 24 h) or on stainless steel (SS) and high-density polyethylene (HDPE) coupons (9 logs CFU/cm^2^; 7 h), followed by phage treatment. Biofilm disruption was measured in vitro at 0, 3, and 6 h as a change in optical density (A_595_). Coupons were treated with STEC serotype-specific phage-cocktails or a 21-phage cocktail (3 phages/serotype) for 0, 3, 6, and 16 h, and surviving STEC populations were enumerated. (3) Results: Of the 52 phages, 77% showed STEC biofilm disruption in vitro. Serotype-specific phage treatments reduced pathogen population within the biofilms by 1.9–4.1 and 2.3–5.6 logs CFU/cm^2^, while the 21-phage cocktail reduced it by 4.0 and 4.8 logs CFU/cm^2^ on SS and HDPE, respectively. (4) Conclusions: Bacteriophages can be used to reduce STEC and their biofilms.

## 1. Introduction

Shiga-toxigenic *Escherichia coli* (STEC) are human pathogens responsible for multiple foodborne disease outbreaks. Their infections can range from mild to severe bloody diarrhea and hemorrhagic colitis (HC), leading to life-threatening complications, such as hemolytic-uremic syndrome (HUS) and thrombotic thrombocytopenic purpura [1]. While the STEC serotype, O157:H7, is frequently associated with HC and HUS, other non-O157 serotypes, reported to cause similar infections, have also emerged. These include the top six serotypes, O26, O45, O103, O111, O121, and O145, that have been responsible for multiple outbreaks and infections in the last two decades [2,3,4]. According to the Centers for Disease Control and Prevention (CDC) estimates, STEC O157 and non-O157 together cause more than 265,000 illnesses each year in the United States [5]. These infections could be due to the consumption of contaminated fresh produce, undercooked meat, unpasteurized milk, or drinking water [6]. Other factors include person to person contact, livestock handling and contact with their environment on the farm, and livestock events or petting zoos [7]. 

These pathogens can also adapt to adverse environmental conditions by forming biofilms on a wide variety of food-contact surfaces, as well as on fresh produce and meat products [8,9,10]. Biofilms consist of a network of adhesive carbohydrates, exopolysaccharides (EPS), that are difficult to penetrate and can protect the bacteria from stressful conditions and antimicrobials [11,12]. It has been reported that STEC biofilms have an increased tolerance to common sanitizers such as chlorine and quaternary ammonium compounds [10,13,14,15]. This could be due to a combination of bacterial resistance mechanisms such as diffusional resistance of the EPS matrix, chemical and enzymatic inactivation of sanitizers and disinfectants, physiological changes in the cell, and induction of stress responses in the cell [16]. The ability of these pathogens to form strong biofilms poses a significant threat of cross-contamination to the food-processing industry. It is therefore critical to develop effective strategies to prevent, remove, or control biofilms in the food industry for improved food safety. 

Bacteriophages are viruses that infect and kill bacteria and have garnered significant attention as antibacterial agents, primarily due to their target-specificity towards the host bacteria [17,18,19]. They are present as commensal microflora in the gastrointestinal tract of animals and have also been isolated from various food and water sources [20,21]. The host specificity and ubiquitous nature of lytic bacteriophages makes them highly desirable as antibacterial agents [18]. Under in vitro conditions, virulent bacteriophages have shown the potential for selective elimination of foodborne pathogens [22,23]. Attention has also been given to the use of bacteriophages for removal of biofilms formed by foodborne pathogens on various food-contact surfaces [24,25,26,27]. Viazis et al. [27] tested a phage cocktail against *E. coli* O157:H7 biofilms on stainless steel (SS), ceramic tile, and high-density polyethylene (HDPE) coupons. The phage cocktail effectively reduced pathogen populations in biofilms on all three surfaces within an hour of the treatment. In another study, phage treatment on spinach-harvesting blades reduced *E. coli* O157:H7 in biofilms by 4.5 logs CFU/blade [25]. The majority of these studies have been limited to evaluating bacteriophage potential against *E. coli* O157:H7 and its biofilms. Very little is known about the effectiveness and applicability of phages against non-O157 STEC and their biofilms. The objective of the present study was to evaluate the efficacy of previously isolated and characterized bacteriophages [28,29] against O157 and non-O157 STEC biofilms in vitro and on food-contact surfaces. 

## 2. Materials and Methods

### 2.1. Bacterial Cultures and Phages

The STEC isolates used in the study included *E. coli* O157:H7 (ATCC 43895, wild type (WT): LF4, KF10), O26 (CDC 2003–3014, WT: QF6, BF8), O45 (CDC 2000–3039, WT: EF2, AF1), O103 (CDC 2006–3008, WT: GF6, AF10), O111 (CDC 2010C-3114, ATCC: 2440, 2180), O121 (CDC 2002–3211, ATCC: 2219, 2203), and O145 (CDC 99–3311, ATCC: 2208, 1652). The WT isolates were retrieved from the Jaroni laboratory culture collection, originally isolated from bovine feces or cattle farm environment [30]. Biofilm-forming capability of these isolates was previously tested in vitro [31]. 

For in vitro studies, overnight cultures of individual STEC isolates were prepared in Luria Broth (LB; MP Biomedicals, CA) and incubated while shaking (180 rpm; V.390 W, Fisher Scientific, NJ) at 37 °C for 16 h. For phage propagation and food-contact surface studies, overnight cultures of individual STEC isolates were prepared in tryptic soy broth (TSB; Bacto™, Difco, BD, Sparks, MD, USA) and incubated statically at 37 °C for 18 h. Where needed, bacterial cocktail suspensions (10^9^ CFU/mL) were prepared from overnight cultures of strains of the same serotypes (Table 1, Table 2 and Table 3) by mixing equal volumes (1:1) and vortexing. Previously isolated and characterized phages (*n* = 52) were used individually or in cocktails to test their efficacy against their host STEC biofilms [28,29]. All the phages were identified as lytic phages belonging to the Myoviridae, Siphoviridae, or Tectiviridae family [28,29].

### 2.2. Phage Preparation

Phages were propagated by suspending 100 µL of overnight culture of host bacterium in molten (0.75%) NZCYM NZ Amine Casamino Acids Yeast Extract MgSO_4_, NaCl (NZCYM) agar (RPI Corp, Mt. Prospect, IL, USA; Fisher Scientific, Waltham, MA, USA) and plating via the double-layer agar method [28]. Phage working-stock solutions were prepared as previously described [28] and stored at 4 °C until further use. Prior to an experiment, phage titers were determined as plaque-forming units per ml (PFU/mL) by serially diluting the phage working stock in phosphate-buffered saline (PBS: pH 7.4: NaCl, KCl, NaH_2_PO_4_ and KH_2_PO_4_, Sigma-Aldrich, St. Louis, MO, USA) and performing a plaque assay [28]. All phage treatments were prepared at a population of 10^9^ PFU/mL.

### 2.3. STEC Biofilm Disruption

#### 2.3.1. In Vitro Biofilm Disruption

In vitro STEC biofilm disruption was determined, as previously described [32], in the following three experiments (I, II, III). In Experiment I, previously isolated and characterized phages (*n* = 52) were used individually to test their efficacy against their host STEC biofilms [28,29]. Phages were isolated from cattle farm environment (water and bovine feces) and were specific to STEC O157:H7, O26, O45, O103, O111, O121, or O145 [28,29]. Based on the results from Experiment I, phages were selected to prepare host-specific cocktails, which were then tested in Experiment II (Table 1). In Experiment III, a multi-phage cocktail, containing 21 phages (3 phages per STEC serotype), was tested against biofilms of multi-serotype STEC cocktail (Table 2).

Overnight pathogen cultures were diluted (1:100) in M9 medium (MP Biomedicals, Irvine, CA), supplemented with 0.4% (wt/vol) glucose (Fisher Scientific, NJ) and minerals (1.16 mM MgSO_4_, 2 µM FeCl_3_, 8 µM CaCl_2_, and 16 µM MnCl_2_; Fisher Scientific, Waltham, MA, USA), and incubated with shaking (180 rpm) for 24 h at 37 °C. Following incubation, bacterial cultures were further diluted (1:100) in M9 medium (containing glucose and minerals) and allowed to form biofilms in 96-well micro-titer plates (Thermo Scientific, Waltham, MA, USA) by aliquoting 150 µL in each well (in triplicates) and incubating the plates at 37 °C for 24 h. Wells filled with sterile M9 were used as the negative control. After incubation, the liquid culture was carefully removed using a micropipette, and the wells were washed three times with PBS (150 µL) without disturbing any biofilm formed at the bottom of the wells. Plates were dried at 37 °C for 15 min, and 150 µL of the respective bacteriophage treatment (individual or cocktail) or PBS (positive control) was added to the wells. Plates were incubated further at 37 °C for 0, 3, and 6 h. After each incubation period, the treatment solution was removed and the wells were washed with PBS, as described above. After drying at 37 °C for 15 min, biofilms were stained by placing crystal violet (CV; Fisher Scientific, NJ) solution (0.1% in distilled water; wt/vol) in the wells for 2 min, washing 3 times with PBS, and drying (37 °C for 15 min). The stain was released with 150 µL of ethanol:acetone solution (80:20; vol/vol; Fisher Scientific, NJ), and biofilm disruption was quantified by measuring the optical density (OD) at 595 nm (SpectroMax M3; Molecular Devices LLC., San Jose, CA). Wells filled with ethanol:acetone solution were used as the blank. 

#### 2.3.2. STEC Biofilm Disruption on Food-Contact Surfaces

The efficacy of bacteriophages to disrupt STEC biofilms on food-contact surfaces was studied in two experiments (I and II). In Experiment I, a cocktail of each STEC serotype (described in 2.1) was used to form biofilms on stainless steel (SS) and high-density polyethylene (HDPE) coupons and treated with their respective phage cocktail for 16 h (Table 3). 

In Experiment II, a 21-phage cocktail was evaluated against biofilms formed by a cocktail of 14 STEC isolates (2 isolates per serotype) on SS and HDPE coupons for 0, 3, 6 and 16 h (Table 2).

##### Preparation of Coupons 

Stainless steel (304 finish, type 4; Stillwater Steels, Stillwater, OK, USA) and HDPE (1/8” × 24” × 48”; Polymersan, Hialeah, FL, USA) coupons (2 × 5 cm^2^) were used for the study and prepared as described by Hood and Zottola [33]. Coupons were cleaned by soaking in acetone (30 min), followed by distilled water rinse (5 min), soaking (1 h) in 1 N NaOH (Fisher Scientific, Waltham, MA, USA) and sonicating (40 KHz; Branson, CT, USA) in distilled water (1 h). Following sonication, coupons were rinsed in distilled water, air-dried, and sterilized prior to use. 

##### Phage Treatment of STEC Biofilms 

For each STEC serotype, three coupons were inoculated, where one coupon was designated as the inoculated, untreated control, one was treated with the respective phage cocktail, and one treated with PBS (control). In each experiment (I and II), coupons were first immersed in 30 mL suspension of bacterial cocktail (10^9^ CFU/mL) in a 50 mL centrifuge tube (Fisher Scientific, GA) and incubated at 25 °C for 2.5 h to facilitate bacterial attachment to the surface. Coupons were gently removed from the suspension using sterile forceps and placed in a sterile 50 mL tube for 5 h in a biosafety hood, allowing further bacterial attachment and biofilm formation. After incubation, coupons were rinsed in 30 mL of sterile distilled water to remove unattached bacterial cells from the surface. One coupon of each surface material (SS or HDPE) was sampled to determine the initial pathogen population within the biofilm (inoculated, untreated control). The remaining inoculated coupons were then suspended in 30 mL of bacteriophage (10^9^ PFU/mL) treatment (Table 3) or PBS at 37 °C for 16 h in Experiment I, and 0, 3, 6, and 16 h in Experiment II. After incubation, coupons were sonicated for 5 min at 40 KHz to dislodge bacterial cells from the coupon surface. Immediately following, 3 g of glass beads (4 mm; Genlantis Diagnostics, CA) were added to the tube and agitated for 1 min using a vortex (Fisher Scientific, NJ) to remove any remaining attached cells from the coupon [26]. Surviving pathogen populations in the solution were enumerated (CFU/cm^2^) on tryptic soy agar (TSA, Fisher Scientific, NJ) or STEC CHROM agar (CHROMagar, Paris, France). Any injured cells were recovered by 24 h enrichment in TSB and plating on TSA at 37 °C. 

### 2.4. Statistical Analysis

All the experiments were repeated three times. Surviving STEC populations, recovered after treatments, were converted to log_10_ CFU/cm^2^, and the mean values of the three replicates were obtained. Data were analyzed using the General Linear Model (SAS v.9.3 software; SAS Inst., Cary, NC, USA) to determine analysis of variance (ANOVA) for phage treatment effects. Significant differences between the treatment means were separated by the least significant difference (LSD) at *p* < 0.05.

## 3. Results and Discussion

### 3.1. STEC Biofilm Disruption 

Effectiveness of individual phages and cocktails of phages to disrupt biofilms of seven STEC serotypes, in vitro and on food-contact surfaces (SS and HDPE), was determined. Results demonstrated that the phages were more effective in reducing biofilms on food-contact surfaces than in vitro. 

#### In Vitro STEC Biofilm Disruption 

Biofilm disruption by individual serotype-specific bacteriophages was determined in vitro by measuring OD (595 nm) at 0, 3, and 6 h. Among the phages tested (*n* = 52), 77% showed significant (*p* < 0.05) reductions in biofilm formation by their respective host bacteria at 3 and 6 h (Figure 1A–D). STEC biofilm disruption was observed, with a reduction in OD from 2.262 nm (0 h) to 0.808 nm (6 h) in these phage-treated wells. 

In Experiment I, where individual phages were tested against their host pathogens, varying results were obtained. All the O157 phages (P1-O157 to P7-O157) showed a reduction in pathogen biofilms at 0 h (A_595_ = 0.472–0.696), 3 h (A_595_ = 0.284–0.441), and 6 h (A_595_ = 0.202–0.321), except phages P3, P5, and P7, where an increase in absorbance was observed at 6 h (Figure 1A). At 3 and 6 h, phage P4-O157 showed the highest reduction compared to the rest of the phages. Non-O157 STEC biofilms, treated with phages, also showed significant (*p* < 0.05) reductions at 0 h (A_595_ =1.034–3.853), 3 h (A_595_ = 0.506–2.631), and 6 h (A_595_ = 0.202–3.577) (Figure 1A–D). However, STEC O45-specific phages were more effective after 3 h than after 6 h treatment. For some STEC strains, a higher reduction was observed with control treatments, which could be due to the natural phenomenon of decreased growth rate or bacterial cell death after 3 or 6 h in PBS treatment. A list of the most effective serotype-specific phages is provided in Table 4.

Based on the results from Experiment I, selected phage cocktails were tested against their respective STEC serotype biofilms (Experiment II). A list of the most effective serotype-specific phage cocktails is provided in Table 4. Results revealed that all O157-phage cocktails (CT-1 to CT-4) were able to disrupt *E. coli* O157:H7 biofilms, reducing OD from 1.990–2.329 at 0 h to 0.576–0.636 at 6 h (Figure 2A). The CT-3 cocktail showed the highest reduction, from 2.250 at 0 h to 0.748 at 3 h, and CT-2 showed the highest reduction at 6 h, from 2.319 to 0.576. Among the STEC non-O157 phage cocktails, CT5-O26, CT7-O103, and CT12-O145 performed the best by continuing to reduce biofilms at 6 h (Figure 2B,C). Phages from Experiment II were selected to make a 21-phage cocktail (3 phages per serotype) to treat multi-serotype STEC biofilms (2 bacterial isolates per serotype). Results showed that the 21-phage cocktail reduced biofilm after 3 h of treatment, where OD decreased from 2.561 at 0 h to 1.321 at 3 h (Figure 2D). However, a slight increase in absorbance (2.186) was recorded after 6 h of the phage-cocktail treatment (Figure 2D).

In the current study, a slight increase in OD was observed with some phages and phage cocktails at 6 h of treatment. These results are in agreement with other studies assessing phage treatment effects on bacterial biofilms [34,35]. Chan et al. [34] observed an increase in *E. coli* biofilms treated with T4 phage on day 4 after an initial decrease (30%) on day 3. Similarly, Hughes et al. [36] obtained a maximum reduction in biofilm within 2 h or 5 h, depending on the biofilm studied. Factors such as bacterial appendages used by the bacteria for biofilm attachment could decrease the efficacy of phages in the biofilm [36,37]. Additionally, changes in bacterial biofilm profile over time, such as inconsistent expression and accumulation of protein through biofilm development stages, could affect phage efficacy [38,39]. Higher protein production at later stages of biofilm formation could impair phage movement within the medium and interfere with phage efficacy against bacterial biofilm [34,40]. Furthermore, static biofilms are known to produce higher polysaccharide and protein content, which could interfere with phage infection efficacies [41]. Vogeleer et al. [32] showed variation in biofilm matrices of individual non-O157 STEC serotypes under static and dynamic conditions. This variation in biofilm matrix could minimize phage effectiveness against biofilm due to the highly specific nature of its depolymerase enzyme [24,36]. This enzyme is highly specific to EPS produced by host bacteria, and even a minor change in the EPS composition could prohibit its activity, leading to reduced phage activity against biofilm.

Loss in phage effectiveness after 6 h could also be due to generation of phage-resistant mutants [42]. Incubation of biofilms at optimum temperatures (30–37 °C) could trigger the rapid growth of bacteria in the biofilm, including phage-resistant bacteria, resulting in decreased efficacy of the phage treatment [37,43]. It has also been shown that STEC serotypes, grown together to form biofilms, could generate an abundance of morphological variants, which could exhibit varying susceptibility to treatments [32,44]. Some newly generated mutants could lose the receptor responsible for bacteriophage susceptibility and become resistant to phages. The use of more diverse phages with the ability to bind to different bacterial receptors could help in controlling the emergence of bacterial mutants by exerting selective pressure on bacterial populations in the biofilm and increasing the effectiveness of phage cocktail treatment [45]. At the same time, phage resistance has been shown to be transient in bacterial cells, and they could revert to a phage-susceptible state [46]. Additionally, this decrease in phage effectiveness can be overcome by application of a cocktail of multiple phages [46,47]. Multi-phage cocktails were therefore evaluated in the current study for their effectiveness against STEC biofilms. 

### 3.2. STEC Biofilm Disruption on Food-Contact Surfaces

Based on the in vitro experiments, the most effective phage cocktails were selected to conduct application studies on SS and HDPE coupons. In Experiment I, where STEC serotype-specific cocktails were used to form biofilms on SS and HDPE coupons, phage cocktails significantly (*p* < 0.05) reduced bacterial population in the biofilm when compared to the control (Figure 3). Populations of *E. coli* O45, O111, O121, and O145 on SS were reduced to undetectable levels after treatment with respective phage cocktails (Figure 3). A reduction of 3.1 and 3.7 logs was observed in *E. coli* O26 and O103 populations on SS coupons treated with phages compared to the control (Figure 4). On the HDPE surface, *E. coli* O121 population was reduced to undetectable levels after respective phage cocktail treatment (Figure 3). A reduction between 2.3 and 5.3 logs was observed in *E. coli* O26, O45, O103, O111, and O145 populations on HDPE coupons treated with respective phage cocktails compared to the control (Figure 4). No injured cells were recovered from enrichment in TSB and plating on TSA. Phage cocktails for specific STEC serotypes were very effective in reducing pathogen populations on both SS (1.9–4.1 logs CFU/cm^2^ reduction) and HDPE (2.3–5.6 logs CFU/cm^2^ reduction) surfaces. These cocktails were more effective on SS than on HDPE, with the phage cocktails for *E. coli* O45, O145, O111, and O121 reducing pathogen populations to undetectable levels. These differences could be due to the surface type or difference in the biofilm composition, where different types of EPS were produced, which could provide increased resistance to bacteria against phages [47,48,49]. 

In Experiment II, biofilms of a cocktail of 14 STEC isolates (2 isolates per serotype) on SS and HDPE coupons were treated with a 21-phage cocktail (3 phages per serotype). The cocktail showed an immediate (0 h) reduction in STEC populations on HDPE (1.5 logs CFU/cm^2^) and SS (1.0 log CFU/cm^2^) (Figure 4). At 3, 6, and 16 h, the 21-phage cocktail showed significant reductions (*p* < 0.05) in STEC populations on both the surfaces when compared to the control (Figure 4). At 3 h, pathogen populations were reduced by 2.8 and 1.7 logs CFU/cm^2^ on SS and HDPE, respectively. At 6 h, pathogen populations were reduced to undetectable levels on SS and by 1.7 logs CFU/cm^2^ on HDPE. At 16 h, STEC populations were reduced to undetectable levels on both surfaces. These results suggest that the tested bacteriophage treatments effectively reduced STEC populations in the biofilms on SS and HDPE surfaces. 

Studies have previously shown that bacteriophages can effectively reduce pathogens attached to hard surfaces found in food processing environments [26,50,51]. Reductions between 3.5 and 5.4 logs CFU in attached *Listeria monocytogenes* population were observed on SS after 24 h of phage P100 treatment [52]. In a study by Sharma et al. [26], lytic bacteriophages reduced *E. coli* O157:H7 populations by 1.2 logs CFU on SS coupons. In the current study, phage treatments reduced *E. coli* O157:H7 by 1.9 logs CFU/cm^2^ and non-O157 STEC by 3.1–4.1 logs CFU/cm^2^ on SS surfaces. Studies have shown that phages can diffuse through the biofilm formed by bacteria [53] and that the formation of biofilms does not provide additional protection to bacteria against phage attack [26]. The biofilm break-down mechanism of phages is speculated to be associated with enzymatic means. Studies have shown that phages produce enzymes that can degrade the EPS layer, the major component of a biofilm matrix [24,36]. Hughes et al. [36] showed that bacteriophages specific for *Enterobacter agglomerans* disrupted biofilm through a combination of lytic activity against bacterial cells and degradation of EPS through phage-associated polysaccharide depolymerase enzyme. The presence of phage-depolymerase in an O45-phage (*p*-9), used in the current study, has been confirmed (data not shown), suggesting that other phages in the cocktail tested may also produce EPS-degrading enzyme, resulting in reduction of bacteria embedded in the biofilm matrix. 

Variations in pathogen attachment and biofilm formation were observed on the food- contact surfaces tested in the present study. Higher STEC attachment was observed on HDPE (up to 5.6 logs CFU/cm^2^) compared to SS (up to 4.3 logs CFU/cm^2^). A similar trend has been observed in previous studies. Higher bacterial attachment and biofilm formation has been observed on plastic surfaces (polyethylene, polypropylene, polyvinyl chloride) than on stainless steel or glass surfaces [48]. Variations found in polymer surfaces, such as smoothness, charge, Zeta-potential, and active chemical groups, could influence bacterial attachment to the surface [54]. Studies have also shown that a variety of active chemical groups released by HDPE surfaces, such as phenols, quinones, aromatic hydrocarbons, aldehydes, and ketones, could be utilized by bacteria as carbon sources and result in higher bacterial attachment [55,56,57].

## 4. Conclusions

This study has shown the potential for the use of lytic bacteriophages against seven STEC serotypes as a treatment to control biofilm formation. Future studies need to be conducted to understand phage–bacteria interactions in biofilm using kinetic modeling. The preliminary results from the current study show promise due to the fact that phages within the biofilms effectively targeted and lysed STEC and were also able to disperse the extracellular matrix forming the biofilm. 

## Figures and Tables

**Figure 1 foods-12-02787-f001:**
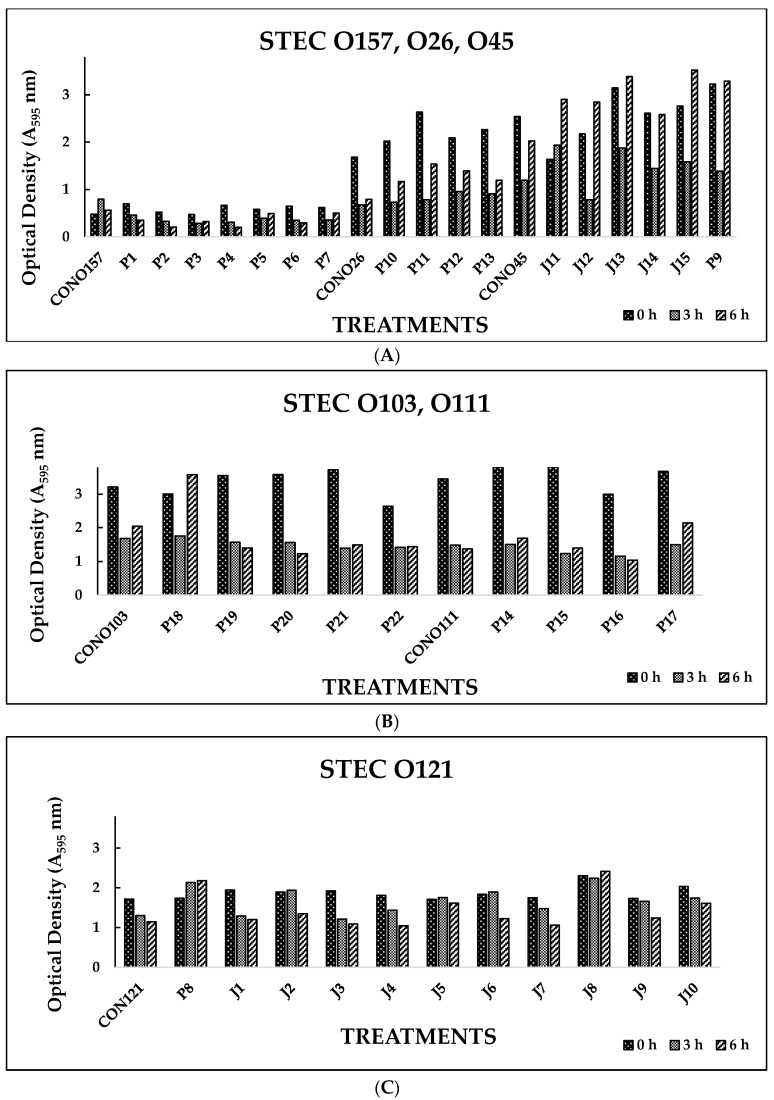
(**A**). In vitro biofilm disruption by individual bacteriophages specific to *E. coli* O157:H7, O26, and O45 after treatment for 0, 3, and 6 h. (**B**). In vitro biofilm disruption by individual bacteriophages specific to *E. coli* O103 and O111 after treatment for 0, 3, and 6 h. (**C**). In vitro biofilm disruption by individual bacteriophages specific to *E. coli* O121 after treatment for 0, 3, and 6 h. (**D**). In vitro biofilm disruption by individual bacteriophages specific to *E. coli* O145 after treatment for 0, 3, and 6 h.

**Figure 2 foods-12-02787-f002:**
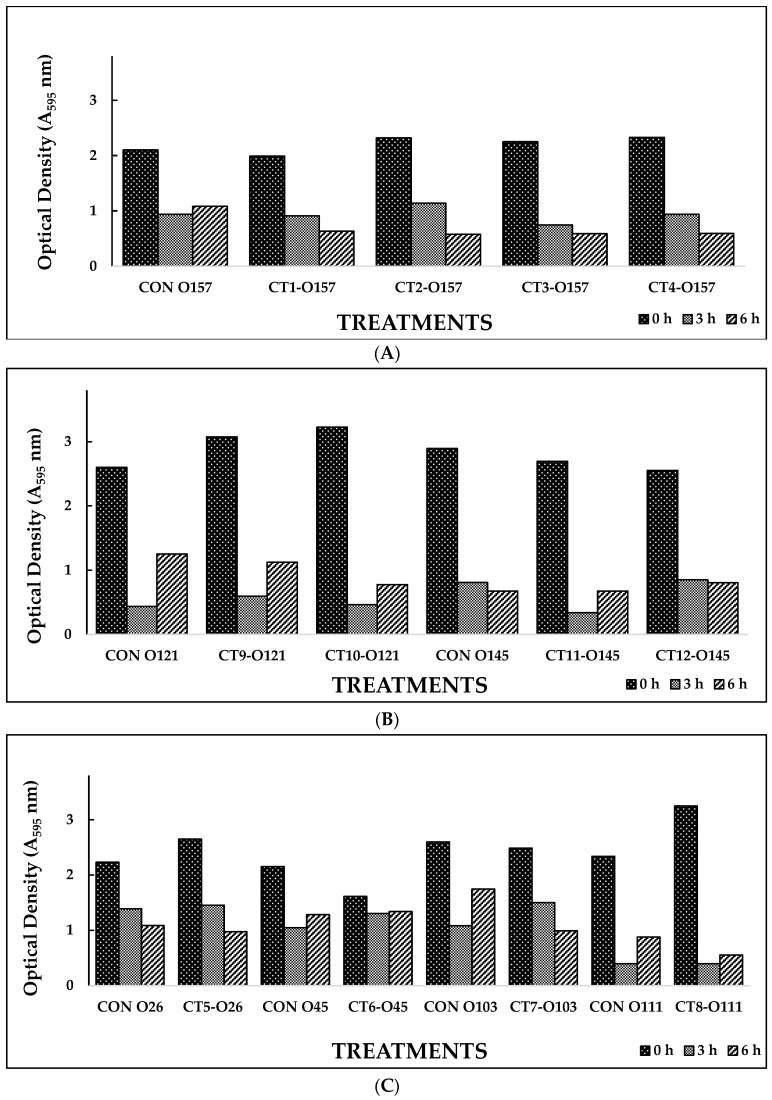
(**A**). In vitro biofilm disruption by bacteriophage cocktails specific to *E. coli* O157 after treatment for 0, 3, and 6 h. (**B**). In vitro biofilm disruption by bacteriophage cocktails specific to *E. coli* O121 and O145 after treatment for 0, 3, and 6 h. (**C**). In vitro biofilm disruption by bacteriophage cocktails specific to *E. coli* O26, O45, O103, and O111 after treatment for 0, 3, and 6 h. (**D**). In vitro STEC biofilm disruption by 21-phage cocktail after treatment for 0, 3, and 6 h. Values represent the average of three replications.

**Figure 3 foods-12-02787-f003:**
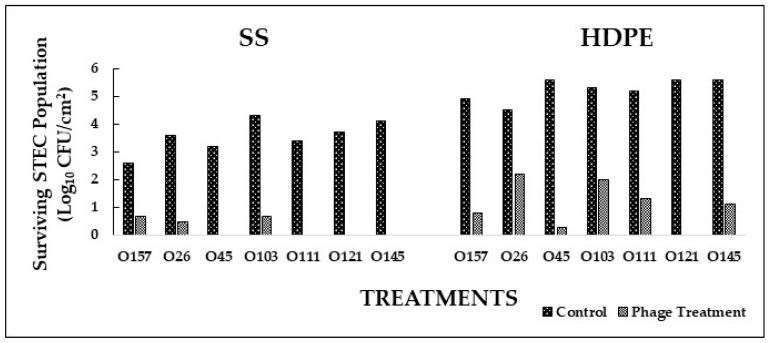
Biofilm disruption on SS and HDPE surfaces by serotype-specific bacteriophage cocktails after treatment for 16 h.

**Figure 4 foods-12-02787-f004:**
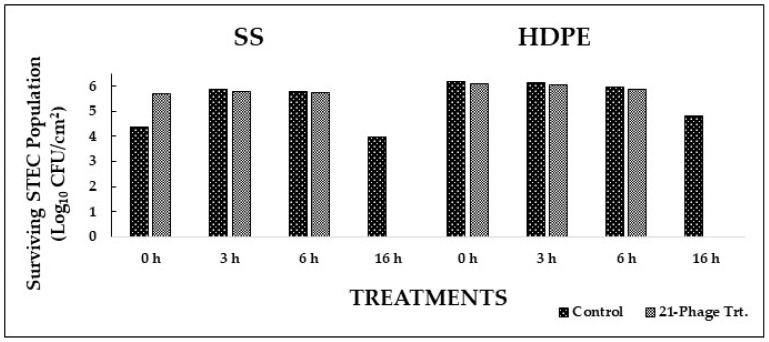
Biofilm disruption on SS and HDPE surfaces by 21-phage cocktail after treatment for 0, 3, 6, and 16 h.

**Table 1 foods-12-02787-t001:** Bacteriophages and STEC isolates used in in vitro Experiments I and II.

STEC	Bacterial Cocktail	Phage Cocktail Treatment (CT)	
O157	ATCC 43895, WT: LF4, KF10	CT1-O157	P1, P2, P4, P6
		CT2-O157	P1, P2, P5, P7
		CT3-O157	P3, P5, P7
		CT4-O157	P2, P3, P4, P7
O26	CDC 2003-3014, WT: BF8, QF6	CT5-O26	P10, P11, P12, P13
O45	CDC 2000-3039, WT: AF1, EF2	CT6-O45	P9, J12, J13, J15
O103	CDC 2006-3008, WT: AF10, GF6	CT7-O103	P19, P20, P21
O111	CDC 2010-3114, ATCC: 2440, 2180	CT8-O111	P14, P15, P16, P17
O121	CDC 2002-3211, ATCC 2219, 2203	CT9-O121	P8, J1, J4, J7
		CT10-O121	P8, J3, J6, J9
O145	CDC 99-3311, ATCC 1652, 2208	CT11-O145	J21, J24, J26, J27
		CT12-O145	J25, J28, J29, J30

**Table 2 foods-12-02787-t002:** Multi-serotype bacterial (14 strains) and phage (21 serotype-specific phages) cocktails used in in vitro Experiment III and in food-contact surface Experiment II.

STEC	Bacterial Cocktail	Phage Cocktail
O157	ATCC 43895, WT LF4	P2, P6, P7
O26	CDC 2003-3014, WT QF6	P11, P12, P13
O45	CDC 2000-3039, WT AF1	P9, J12, J15
O103	CDC 2006-3008, WT AF10	P19, P20, P21
O111	CDC 2010-3114, ATCC 2180	P14, P15, P17
O121	CDC 2002-3211, ATCC 2219	P8, J3, J7
O145	CDC 99-3311, ATCC 1652	J18, J21, J29

**Table 3 foods-12-02787-t003:** Selected phage cocktails used in the food-contact surface study in Experiment I.

STEC	Bacterial Cocktail	Phage Cocktail
O157	ATCC 43895, WT: LF4, KF10	P3, P5, P7
O26	CDC 2003-3014, WT: BF8, QF6	P10, P11, P12, P13
O45	CDC 2000-3039, WT: AF1, EF2	P9, J12, J13, J15
O103	CDC 2006-3008, WT: AF10, GF6	P19, P20, P21
O111	CDC 2010-3114, ATCC: 2440, 2180	P14, P15, P16, P17
O121	CDC 2002-3211, ATCC: 2219, 2203	P8, J3, J6, J9
O145	CDC 99-3311, ATCC: 2208, 1652	J21, J24, J26, J27

**Table 4 foods-12-02787-t004:** Most effective serotype-specific individual phages and phage cocktail treatments in vitro.

STEC	Individual Phage Treatment	Phage Cocktail Treatment (CT)
O157	P4	CT-2, CT-3
O26	P11	CT-5
O45	P9	CT-6
O103	P21	CT-7
O111	P15	CT-8
O121	J3	CT-10
O145	J29	CT-12

## Data Availability

Data are contained within the article.

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
