# Peer review of "Effectiveness of Bacteriophages against Biofilm-Forming Shiga-Toxigenic Escherichia coli In Vitro and on Food-Contact Surfaces"

_foods, 2023, doi:10.3390/foods12142787_

Round 1

Reviewer 1 Report

This study examined the efficacy of a phage cocktail against STEC biofilms in media and on food contact surfaces. The authors tested various phages on STEC of a variety of serotypes. The study is interesting and generally well-written. The manuscript could be improved by addressing the following issues:

1. Line 104 2.3.1 For those previously isolated phages, could the authors briefly mention their sources and genus or species specificity?

2. Line 91 Is it more appropriate to put Experiment I in Table 1 rather than in Table 3?

3. Line 126 Washing trice with PBS?

4. Line 182 Does OD have an unit of nm?

5. Fig 1.2.3, The author are suggested to integrate those separate figures (Fig1a Fig1b) to one multi-panel figure with one figure legend, rather than using separate figure legends.

6. Line 291 Multiphage cocktail used in this study seems not to successfully overcome decreased efficacy as shown in Fig 2D.

7. Fig 2D  The width of columns in this figure are too large compared to other figures. Please revise if possible.

8. For the statistical analysis, the authors used different criteria for significant difference: P<0.10 for in-vitro studies and P<0.05 for food-contact surface studies. Please give some explanation for this. 

Author Response

The authors would like to thank Reviewer 1 for their constructive comments and in helping us make the manuscript better. Please see the responses to your review below. Changes in the revised manuscript are indicated in red font.

  1. Line 104 2.3.1 For those previously isolated phages, could the authors briefly mention their sources and genus or species specificity?

Response: More information has been provided (Lines 106-107) along with authors’ previously published citations where more details on host-specificity and isolation procedures have been discussed [28, 29 in manuscript].

  1. Line 91 Is it more appropriate to put Experiment I in Table 1 rather than in Table 3?

Response: Two studies were done – in-vitro and food-contact. The “in-vitro study” is discussed first in the manuscript and had three experiments (I, II, and III) . Table 1 represents Experiments I and II of the “in vitro study”, while Table 3 represents Experiment-I of the “food-contact study”, which follows the in-vitro study. For this reason, they are assigned the tables, respectively.

  1. Line 126 Washing trice with PBS?

Response: The authors are seeing the word “thrice”. However, it has been changed to “3 times” for clarity (Line 130).

  1. Line 182 Does OD have an unit of nm?

Response: The sentence has been changed to say “OD (595 nm)”.

  1. Fig 1.2.3, The author are suggested to integrate those separate figures (Fig1a Fig1b) to one multi-panel figure with one figure legend, rather than using separate figure legends.

Response: Some of the figures have been integrated (Fig. 1A - STEC O157, O26 and O45; Fig 1B – STEC O111 and O103). Figures 1C and 1D are too large in themselves to be integrated and have been kept separate. The authors have used the journal template for this purpose. 

  1. Line 291 Multiphage cocktail used in this study seems not to successfully overcome decreased efficacy as shown in Fig 2D.

Previously mentioned (Line 225) and an explanation provided (Lines 254-286)

  1. Fig 2D  The width of columns in this figure are too large compared to other figures. Please revise if possible.

Response: Revised

  1. For the statistical analysis, the authors used different criteria for significant difference: P<0.10 for in-vitro studies and P<0.05 for food-contact surface studies. Please give some explanation for this.

Response: This is an oversight on our part. The data was analyzed at one P-value (<0.05) and has been corrected.

Reviewer 2 Report

Effectiveness of Bacteriophages Against Biofilm-forming 2 Shiga-toxigenic Escherichia coli In-vitro and on Food-Contact Surfaces

The topic of using bacteriophages as a biocontrol agent in the food industry is of great interest. The authors present a study in which the potential of different bacteriophages against STEC isolates and its biofilms was investigated. Different STEC including E. coli O157:H7 as well as non-O157 were used. The authors used serotype specific bacteriophages and applied them individual as well as in mixtures of different combinations. The reduction of biofilms was investigated in-vitro and on food-contact surfaces. The results of the study demonstrate that especially the phage cocktails can disrupt STEC biofilms. The authors presented interesting experiments and findings which needs to be improved in some points.

General comment

All in all, this a largely descriptive paper, not taken into account the mechanism, why some phages attack the bacteria quite well with a good elimination and others not. A discussion about the putative mechanisms should be included.

Since similar phages and bacteria have been used in a former study of the authors (doi: 10.1089/phage.2020.0024,the study lacks originality.

Minor comments:
Abstract

Conclusions: One sentence is not enough here.  It should be mentioned that cocktails work better than single strains and/or elimination is serotype-specific. 

Materials and Methods:

The description of the used phages is not sufficient. Please provide more information 
(phage type, etc.) in addition to the citation.

Page 2, line 73: 2.1. Bacterial cultures and phages

The headline does not fit properly since the author also incorporate the tables 1-3 which already deal with the used phage cocktails.

I would recommend to show the tables in the section 2.3. STEC biofilm disruption.

The description of the STEC strains is not sufficient. It would be good to describe the characteristics of the strains. At least, the Shiga toxin types and the origin of the strains should be mentioned. This can also be provided in an additional table.

Page 2, lines 85-87:

Bacterial cocktail suspensions were prepared by mixing (1:1) isolates of STEC serotypes, at a population of 109CFU/ml.

 it does not become clear what was done

 the authors used more than two different isolates per serotype? How was the mixing performed and what was the final cfu number?

Page 2, lines 88:

Why is Table 1 associated with Experiment II?

The order in the Materials and Methods section can be improved. The presentation of the Tables can be improved. Headlines are central but the corresponding text is left.

Page 4, line 121: 

The authors describe in the section “in vitro biofilm disruption” that PBS was used as positive control. I would assume that PBS is somehow the negative control since biofilm will not be disrupted?

Page 5, line 160: 

What was the size of the glass beads? Did the authors check for disruption of cells? The authors mention that injured cells was recovered. What was the results of this? It was not mentioned in the result section.

Results and Discussion

All figures have a bad quality. The figures have different sizes and the y-axes have different scales. This should be adjusted. All in all, the number of figures is too high and the authors should reduce the figures. Probably some figures with individual phages could be kept and the others could be transferred into supplemental data. Moreover, the colouring is not differential enough. The black tones are too similar. Please improve. Another idea is, to combine some figures, because the structure of the figures is similar

3.1.1 In vitro STEC biofilm disruption

For some strains the control shows the same or even a higher reduction in biofilm (Fig.1B, C, D, E, F, G). This is not mentioned in the text. Please also describe these results. It is absolutely essential to describe the phenomenon, that the controls in some cases show a higher reduction in Biofilm disruption as the phage treatment. 

Page 8, line 221 ff.:

The authors describe that all O157-phage cocktails were able to disrupt E. coli biofilms. But this reduction is also shown for the CON O157. Is this the control in which biofilms were treated with PBS?

Page 8, line 225: 

Among the STEC nonO157: please change to STEC non-O157.

Page 8, line 229: 

Name the phage cocktail.

Page 10, line 263: 

In the current study, (…). Results of the current study (…).

Reformulate the sentence.

Page 10, line 263: 

T-4 change to T4 phage

3.2. STEC biofilm disruption on food-contact surfaces

Which population is shown here? The injured cells which were recovered or the population which was plated directly after detaching?

This is not clear here. 

Page 11, line 299, 302: 

Compared to the control. Which control was used? The untreated control or the control with PBS? Why did the authors perform these two controls? 

Page 11, line 310:

What does undetectable levels mean here? What was the detection limit?

Conclusion

Page 13, line 368

This study has shown the potential for the use of (…).

Author Response

The authors would like to thank the reviewer for their constructive comments and helping us make the manuscript better. It is greatly appreciated. The changes in the manuscript are indicated in red font. Please see the responses to your comments below.

General comment

All in all, this a largely descriptive paper, not taken into account the mechanism, why some phages attack the bacteria quite well with a good elimination and others not. A discussion about the putative mechanisms should be included.

Since similar phages and bacteria have been used in a former study of the authors (doi: 10.1089/phage.2020.0024,the study lacks originality.

Response: The study mentioned by the reviewer was done with spinach and cucumbers on limited number of phages. The current study was conducted in-vitro and on food-contact surfaces to test the biofilm disruption capability of several (n=52) phages and their cocktails.

Minor comments:
Abstract: 

Conclusions: One sentence is not enough here.  It should be mentioned that cocktails work better than single strains and/or elimination is serotype-specific. 

Response: The authors were limited by the word count (200 words or less) for the abstract. More details are provided in the conclusion in the manuscript.

Materials and Methods:

The description of the used phages is not sufficient. Please provide more information 
(phage type, etc.) in addition to the citation.

Response: More information about the phages has been added, along with the citation of previous publications by authors where the phages are described in more detail.

Page 2, line 73: 2.1. Bacterial cultures and phages

The headline does not fit properly since the author also incorporate the tables 1-3 which already deal with the used phage cocktails.

Response: Headline has been changed as suggested by the reviewer.

I would recommend to show the tables in the section 2.3. STEC biofilm disruption.

Response: As suggested, tables are now shown in section 2.3 STEC biofilm disruption.

The description of the STEC strains is not sufficient. It would be good to describe the characteristics of the strains. At least, the Shiga toxin types and the origin of the strains should be mentioned. This can also be provided in an additional table.

Response: The origin of the strains has been mentioned.

Page 2, lines 85-87:

Bacterial cocktail suspensions were prepared by mixing (1:1) isolates of STEC serotypes, at a population of 109CFU/ml.

à it does not become clear what was done

Response: More information has been provided for clarity.

à the authors used more than two different isolates per serotype? How was the mixing performed and what was the final cfu number?

Response: Yes, 2 or more isolates per serotype were used. Isolates were mixed by gentle vortexing and the final cfu number was ~109 cfu/ml. This information is provided in Lines 85-88.

Response:

Page 2, lines 88:

Why is Table 1 associated with Experiment II?

Response: Table 1 should be associated with Experiment I and II for in vitro studies. This has been corrected to reflect that. Two studies were done – in vitro and food-contact. The “in vitro study” is discussed first in the manuscript and had three experiments (I, II, and III) . Table 1 represents Experiments I and II of the “in vitro study”.

The order in the Materials and Methods section can be improved. The presentation of the Tables can be improved. Headlines are central but the corresponding text is left.

Response: Corrections have been made throughout.

Page 4, line 121: 

The authors describe in the section “in vitro biofilm disruption” that PBS was used as positive control. I would assume that PBS is somehow the negative control since biofilm will not be disrupted?

Response: The positive control refers to the bacterial cultures that were grown and allowed to form biofilms and then treated with PBS, instead of phage treatments. Negative control was used separately and is mentioned in Line 122.

Page 5, line 160: 

What was the size of the glass beads? Did the authors check for disruption of cells? The authors mention that injured cells was recovered. What was the results of this? It was not mentioned in the result section.

Response: More information has been provided in the Methods and Results section.

Results and Discussion

All figures have a bad quality. The figures have different sizes and the y-axes have different scales. This should be adjusted. All in all, the number of figures is too high and the authors should reduce the figures. Probably some figures with individual phages could be kept and the others could be transferred into supplemental data. Moreover, the colouring is not differential enough. The black tones are too similar. Please improve. Another idea is, to combine some figures, because the structure of the figures is similar

Response: Authors agree with the reviewer. The suggestions have been incorporated in the revised manuscript. Moreover, the figures change their format when converted to PDF. We are only submitting the word file of the revised manuscript for this reason.

3.1.1 In vitro STEC biofilm disruption

For some strains the control shows the same or even a higher reduction in biofilm (Fig.1B, C, D, E, F, G). This is not mentioned in the text. Please also describe these results. It is absolutely essential to describe the phenomenon, that the controls in some cases show a higher reduction in Biofilm disruption as the phage treatment. 

Response: Corrected and addressed in the results and discussion section.

Page 8, line 221 ff.:

The authors describe that all O157-phage cocktails were able to disrupt E. coli biofilms. But this reduction is also shown for the CON O157. Is this the control in which biofilms were treated with PBS?

Response: The sentence has been corrected to incorporate the reviewer’s concern.

Page 8, line 225: 

Among the STEC nonO157: please change to STEC non-O157.

Response: Corrected

Page 8, line 229: 

Name the phage cocktail.

Corrected

Page 10, line 263: 

In the current study, (…). Results of the current study (…).

Reformulate the sentence.

Response: Corrected

Page 10, line 263: 

T-4 change to T4 phage

Response: Corrected

3.2. STEC biofilm disruption on food-contact surfaces

Which population is shown here? The injured cells which were recovered or the population which was plated directly after detaching?

This is not clear here. 

Response: No injured cells were recovered. The population shown is the one enumerated after detaching.

Page 11, line 299, 302: 

Compared to the control. Which control was used? The untreated control or the control with PBS? Why did the authors perform these two controls? 

Response: Authors are referring to the positive control treated with PBS. The untreated control was used to determine the initial population that was inoculated on the coupons.

Page 11, line 310:

What does undetectable levels mean here? What was the detection limit?

Response: The detection limit of the experiments was 0.5 log10 cfu/cm2.

Conclusion

Page 13, line 368

This study has shown the potential for the use of (…).

Response: Corrected